# Microbial Community Structure and Predictive Functional Analysis in Reclaimed Soil with Different Vegetation Types: The Example of the Xiaoyi Mine Waste Dump in Shanxi

**Dong Zhao [1], Huping Hou [2],\*, Haiya Liu [3], Chen Wang [4], Zhongyi Ding [3],\* and Jinting Xiong [5]**

1  Science and Technology and Foreign Exchange and Cooperation Division, Jiangsu Provincial Natural Resources Department, Nanjing 210017, China
2  Engineering Research Center of Ministry of Education for Mine Ecological Restoration, China University of Mining and Technology, Xuzhou 221116, China
3  School of Public Policy and Management, China University of Mining and Technology, Xuzhou 221116, China
4  Observation and Research Station of Ecological Restoration for Chongqing Typical Mining Areas, Ministry of Natural Resources, Chongqing Institute of Geology and Mineral Resources, Chongqing 401120, China
5  School of Environment Science & Spatial Informatics, China University of Mining and Technology, Xuzhou 221116, China
*  Correspondence: houhuping@cumt.edu.cn (H.H.); dingzhongyi@cumt.edu.cn (Z.D.); Tel.: +86-13852144183 (H.H.); +86-13852141910 (Z.D.)

**Abstract:** We explored the characteristics of soil bacterial communities and their ecological functions under different types of vegetation reclamation in open-pit mines on the Loess Plateau, which is the guiding significance for the selection of vegetation and the improvement of restoration effect in mining areas. The research object was to reclaim the soil of the aluminum mine waste dump in Xiaoyi County, Shanxi. The soil characteristics were measured under different types of vegetation reclamation. The soil bacterial community under different vegetation reclamation was measured using the 16S rRNA gene high-throughput sequencing technology. The ecological function was predicted using the PICRUSt method. The correlation between soil physical and chemical properties and bacterial community structure and function was analyzed. From the results, (1) the bacterial compositions of the reclaimed soil samples were 33 phyla, 90 classes, 121 orders, 207 families, 298 genera, and 140 species. The abundance and diversity of the soil microbial community showed the rule of yellow rose > lespedeza and sweet wormwood herb > alfalfa. (2) Proteobacteria were the dominant bacteria in alfalfa and sweet wormwood herb samples, accounting for 36.09–43.36%. Proteobacteria and actinobacteria were the dominant bacteria in the yellow rose and lespedeza samples accounted for 53.34–53.39%. α-Proteobacteria, actinobacteria, and β-proteobacteria were the dominant bacteria of the four vegetation types. The relative abundance of the α-proteobacteria and β-proteobacteria was positively correlated with soil organic carbon (SOC) and negatively correlated with soil total kalium (TK). Actinobacteria were positively correlated with available kalium (AK) and negatively correlated with SOC and total nitrogen (TN). (3) There was no difference in the primary functions of the soil bacterial community after the reclamation of different plants, and the main functions were metabolism, genetic information processing, and environmental information processing, with the function abundance accounting for 81.52%. (4) The abundance of functional genes in the metabolism of other amino acids, folding, sorting, and degradation and glycan biosynthesis and metabolism were relatively rich in the rhizosphere soil of yellow rose. The abundance of functional genes in signal molecules and interaction, transport, and catabolism in the rhizosphere soil of lespedeza was the highest. The abundance of functional genes in carbohydrate metabolism, translation, and energy metabolism in the rhizosphere soil of alfalfa was the highest. Therefore, there were significant differences in the structure and function of rhizosphere soil microbial communities among yellow rose, lespedeza, sweet wormwood herb, and alfalfa, and they were also affected by the soil properties. Hence, we concluded that the differences and diversity of soil microbial structure and function can help select plants for the sustainable development of soil remediation in mining areas.

**Keywords:** soil remediation; different vegetation types; soil bacterial community; high-throughput sequencing; function prediction

## 1. Introduction

The exploitation of open-pit mineral resources has caused serious damage to vegetation, soil, water, and other environments, resulting in landscape fragmentation, a sharp reduction in biodiversity, soil degradation, and other problems. Soil reconstruction and vegetation reconstruction technology is one of the measures used for land reclamation and ecological restoration. However, the change in reclamation vegetation type not only changes the landscape of land, but also changes the material cycle and energy flow of the soil ecosystem, which affects the structure and function of soil microorganisms and the sustainable development of the soil ecosystem after reclamation. Soil microorganism is an important factor for soil ecosystem stability and an important indicator of soil quality change [1–4]. Plants participate in the material cycle and energy flow in the soil system and change the physical and chemical properties of soil and soil bacteria through the input of litter and root exudates, thus forming a feedback system of interaction between soil microorganisms and vegetation [5–7]. Researching the interrelationship among plants, soil microorganisms, and environmental factors, as well as their impact on the stability of biological communities, is the key measure to solving the problem of high-quality soil development after ecological restoration [8].

To date, there have been relevant studies on the effect of different vegetation configurations on soil properties [9–11]. For example, different vegetation configurations lead to different water and soil nutrition reduction effects in the ecosystem [12,13]. The shrub-grass intercropping mode improved the activity of soil microorganisms in the sandy grassland during the restoration process [14]. Artificial reclamation or natural vegetation restoration can significantly improve the soil nutrient conditions, improve the fertility of the reclaimed farmland soil, and improve the soil quality of the mining area. With the increase in the restoration time, the improvement effect is more obvious [15]. The total amount of microbes and percentage of actinomycetes and fungi among microorganisms have been found to gradually increase in reclaimed soil with increased reclamation years [11]. There is a significant correlation between different types of reclaimed plants and soil microbial community structure. When ground plant species gradually decrease and dominant plants degenerate, the number of soil aerobic and anaerobic azotobacters decreases, showing the characteristics of synergistic changes [16–18]. The role of soil microorganisms in the environment is mainly realized through the difference in metabolic functions of microbial communities. Understanding the distribution characteristics of microbial functions plays an important role in better understanding the relationship between plants and soil microbial communities, as well as the response of soil ecosystems to environmental changes [19].

In the past, the focus of soil microbial community analysis was the composition and structure of the soil microbial community. The research on soil bacteria began to change from structure to function recently. Scholars all over the world are committed to the prediction and analysis of soil bacterial function and try to reveal the important role of soil bacteria through the study of soil bacterial function [20–22]. Three age groups of P. sylvestris plantations (25 a, 34 a, and 43 a) were selected to determine soil bacterial community composition and functional groups, the active bacterial metabolism in 43 a plantation was conducive to nutrients absorption and utilization by plants [23]. The microbial community exhibited clear species-specific and spatial differences in their compositions and functional groups [24–27]. Inoculation with Medicago sativa rhizobium could increase amino acid metabolism, which was conducive to plant N-nutrient cycle, and inoculation with arbuscular mycorrhizal (AM) fungi may have a certain inhibitory effect on N-cycle [28,29]. The 16S rRNA high-throughput sequencing can predict the bacterial flora metabolic function spectrum corresponding to the gene sequence through PICRUSt (Phylogenetic Investment

of Communities by Reconstruction of Unobserved States) software [30–33]. Compared with metagenome research, PICRUSt function prediction analysis is more convenient and cheaper, and the prediction effect is more reliable.

The ecological environment of the Loess Plateau in China is fragile. Ecological restoration of the damaged areas of open-pit mining is the key work of governance. Maintaining the sustainable development of the restored ecosystem is the key element of ecological restoration. However, there is little research on the stability and sustainable development of soil ecosystems under different vegetation configuration modes. Based on the above analysis, the characteristics of soil microbial communities under different vegetation restoration were studied in the waste dump of Xiaoyi open aluminum mine pit, Shanxi Province. The structural characteristics and functional sequencing of soil microbial communities were analyzed using the 16S rRNA gene high-throughput sequencing technology and the PICRUSt method. This paper studied from two aspects: (1) analyze the differences of soil microbial communities structural and functional in different types of vegetation restoration; (2) analyze the influence among soil properties, soil microbial communities, and soil functions. These findings provide a theoretical reference for the selection of reclamation plants and the evaluation of reclamation effects.

## 2. Materials and Methods

### 2.1. Study Area

The study area is located in the open-pit mining area of Ke'e Village, Yangquanqu Town, Xiaoyi City, Shanxi Province in the hilly and gully area of the Loess Plateau, with geographic coordinates of 111°30″–111°31″ E and 37°08″–37°09″ N (Figure 1). It belongs to the warm temperate continental semi-arid and semi-humid climate. The average altitude is 1196.89 m, the annual average temperature is 10 °C–12 °C, the annual average sunshine is 2640.7 h, the average total solar radiation is 147 kcal/cm$^2$, the sunshine rate is 60%, the average precipitation is 486 mm, and the annual average frost-free period is 190 days. The soil type is yellow cotton soil, light loam, and medium loam, with a loose structure and good air permeability. The vegetation type mainly comprises sweet wormwood herb (Artemisia annua L.), alfalfa (Medicago sativa Linn), lespedeza (Lespedeza bicolor Turcz), and yellow rose (Rosa xanthina Lindl).

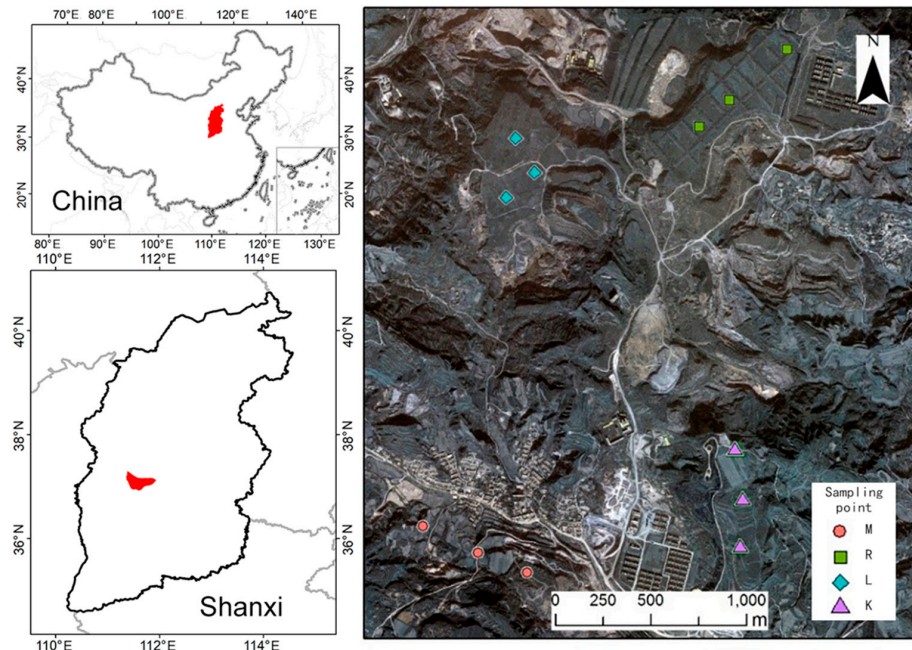

**Figure 1.** Sampling sites in the study area (M is alfalfa; R is yellow rose; L is lespedeza; K is sweet wormwood herb).

The study area is located in the middle and upper part of the lower section of the Upper Carboniferous Benxi Formation. The ore-bearing rock series is composed of mainly iron ore, bauxite, refractory clay ore, and coal line, with a thickness of 8–20 m. The mining area is dominated by open-pit bauxite mining and coal mining. Since 2010, the waste dump has been reclaimed using "raw soil backfilling → 50 cm topsoil covering → land leveling → vegetation planting". The reclaimed soil comes from the surrounding soil and stripped topsoil, and the vegetation types mainly include yellow rose, lespedeza, sweet wormwood herb, and alfalfa.

### 2.2. Soil Collection and Analysis

Sampling was conducted on 11 August 2021, and reclamation of the selected sample plot was carried out on 1 June 2015. After reclamation, the sampled plot was located on the ground platform. Four types of vegetation were selected for the sampled plot, namely, sweet wormwood herb (K), lespedeza (L), yellow rose (R), and alfalfa (M). Each sampled plot was set with five 4 m$^2$ grassland survey sampled points, and five soil samples were collected for mixing at each sampled point. The "five-point method" was used to collect soil samples (0–20 cm soil layer), remove the root system and other debris, fully and evenly mix them, take approximately 300 g of the mixture into a sealed sterile bag, and take 10 g of the mixture into a centrifuge tube, and mark them. The soil samples used for molecular biological analysis were transported to Shanghai Parsono Biotechnology Co., Ltd. (NovaSeq 6000 PE250, Illumina corporation, USA), on dry ice. The soil samples used for determining soil physical and chemical properties were taken back to the laboratory processes, such as natural air drying, grinding, and sieving before preliminary treatment.

The physical and chemical properties of the soil were determined using conventional analysis methods [34]. The semimicro-Kjeldahl method to was used to determine soil total nitrogen (TN), and the HF–HClO$_4$–HNO$_3$ method and inductively coupled plasma atomic emission spectrometer (iCAP 6300 ICP-OES Spectrometer Thermo Scientific, USA) were used to analyze total phosphorus (TP) and total kalium(TK). Organic carbon was measured using the potassium dichromate oxidation–outer heating method, and the soil pH was measured with a 1:5 soil solution using an acidimeter (Sartorius PB-10, Germany). The soil water content (WC) was measured using the GS-1 soil moisture sensor, and the soil temperature was measured using the Delta-T temperature sensor [32].

### 2.3. Soil DNA Extraction and PCR Amplification

The DNA of the soil samples was extracted using the E.Z.N.A.® DNA/RNA Isolation kit. The quality of the DNA extraction was detected using 0.8% agarose gel electrophoresis, and an ultraviolet spectrophotometer was used to quantify the DNA. Primers were designed for the ITS1 and V4–V5 regions of the target fragment, a sample-specific barcode sequence was added, and a Q5 high-fidelity DNA polymerase (NEB) was used for PCR amplification. The reaction system for amplification was preincubated at 98 °C for 30 s (98 °C/10 s, 50 °C/30 s, and 72 °C/30 s) × 35 cycles, with extension at 72 °C for 2 min. The PCR amplification products were detected using 2% agarose gel electrophoresis, and the target fragments were cut and recovered with a gel recovery kit (AXYGEN Company). The Quant-iT PicoGreen dsDNA Assay Kit was used for fluorescence quantification of the PCR amplified and recovered products, and the samples were mixed according to the corresponding proportion [33,34].

### 2.4. 16S rRNA High-Throughput Sequencing

The *16S rRNA* gene of soil bacteria were sequenced within three days of sampling. The DNA of soil samples was extracted using the E.Z.N.A® Soil DNA Kit (Omega Bio-Tek, Norcross, GA, USA) according to the manufacturer's protocols. The V4–V5 area of the *16S rRNA* of soil bacteria was amplified by polymerase chain reaction (PCR). The applied degenerate primers included the 515F 5′-barcode-GTGCCAGCMGCCGCGG-3′ and 907R 5′–CCGTCAATTCMTTT RAGTTT-3′.

The PCR amplification program was as follows: pre-degeneration at 95 °C for 2 min, degeneration at 95 °C for 30 s, annealing at 55 °C for 30 s, and extension at 72 °C for 30 s. The above steps were repeated for 25 cycles, followed by a final extension at 72 °C for 5 min. The PCR reaction was performed in triplicate in 20 μL mixtures, containing 4 μL of 5 × FastPfu buffer solution, 2 μL of 2.5 mM dNTP, 0.8 μL of each primer (5 μM), 0.4 μL of FastPfu polymerase, and 10 ng template DNA. The amplicons were extracted from 2% agarose gel, and they were then purified using the AxyPrep DNA gel extraction kit (Axygen Biosciences, Union City, CA, USA) according to the manufacturer's instructions. Finally, samples were quantified using QuantiFluor $^{TM}$-ST (Promega, WI, USA).

Purified PCR products were quantified using the Qubit$^{®}$3.0 (Promega Corporation, USA) fluorometer, and amplicons with different sequences were mixed evenly. The collected DNA products were then used to establish the Illumina pair-end library with a "Y" shaped connector. An Illumina Nextera$^{®}$ XT Index Kit (Illumina, San Diego, CA, USA) was used to attach dual indices and Illumina sequencing adapters. Following the second PCR, samples were re-cleaned with AMPure XP beads (Beckman Coulter, Pasadena, CA, USA) and quantified. The amplicon library was then pair-end sequenced (2 × 250) using the Illumina MiSeq platform (Shanghai BIOZERON Co., Ltd.) according to the standard scheme. The read raw data were then saved in the National Center for Biotechnology Information (NCBI) sequence read archive (SRA) database (Accession Number: SRP05270).

The UCLUST tool of QIIME software was used to merge and divide the sequences obtained into OTUs with 97% similarity. By comparing the OTU representative sequence with the template sequence of the Silva database (Release115, http://www.arb-silva.de, 11 October 2021), we obtained the taxonomic information corresponding to each OTU. Using Mothur Ver1.21, the richness Chao index, ACE index, and community evenness Shannon index of the soil microbial community were calculated. The samples clustered and generated a tree graph based on the unweighted group average method (UPGMA) of the Unweighted UniFrac distance matrix. PICRUSt software was used for function prediction and analysis, and the function abundance heat map was drawn according to the function abundance.

### 2.5. Statistical Analysis

Redundancy analysis (RDA) was performed based on the linear model using R 3.5.1, and the RDA diagram was drawn. The standard error was calculated, and ANOVA was performed using SPSS 19.0. The significant difference was tested using the LSD multiple comparison method ($p < 0.05$), and Origin 2018 software was used to draw the graphs.

## 3. Results and Analysis
### 3.1. Analysis of the Chemical Properties of Rhizosphere Soil of Different Reclaimed Plants

The soil quality of different plant types for reclamation was determined (Table 1). The results showed that the WC and TP content (26.20% and 0.56 g·kg$^{-1}$, respectively) reached their highest in lespedeza samples, and they were significantly higher than those of other samples. The soil organic carbon (SOC) content of alfalfa samples was significantly higher than that in other samples, and it was 49.38% higher than that in the lespedeza samples. The TN and available kalium contents of yellow rose were the highest, and they were 25.64% and 48.34% higher than those in lespedeza and alfalfa, respectively. There was no significant difference in the TP and pH among the different plant samples. As the water storage capacity of lespedeza in the Loess Plateau is high, and lespedeza is used as an important plant to retain precipitation and improve the soil during the afforestation of barren mountains and wastelands. The SOC of alfalfa is high and related to its high yield, drought tolerance, and barren tolerance.

**Table 1.** Chemical properties of reclaimed soils with different plant types.

| Vegetation Types | WC/% | SOC/g·kg$^{-1}$ | TN/g·kg$^{-1}$ | TP/g·kg$^{-1}$ | TK/g·kg$^{-1}$ | AK/mg·kg$^{-1}$ | pH |
|---|---|---|---|---|---|---|---|
| Yellow rose (R) | 21.70 ± 1.70 b | 4.88 ± 0.57 b | 0.49 ± 0.03 a | 0.42 ± 0.01 c | 19.59 ± 0.78 a | 155.93 ± 29.59 a | 8.26 ± 0.05 a |
| Lespedeza (L) | 26.20 ± 0.70 a | 4.01 ± 1.97 c | 0.39 ± 0.13 b | 0.56 ± 0.01 a | 18.65 ± 0.24 a | 123.23 ± 9.21 b | 8.22 ± 0.04 a |
| Sweet wormwood herb (K) | 21.28 ± 8.56 b | 5.21 ± 3.09 b | 0.41 ± 0.15 b | 0.49 ± 0.12 b | 18.00 ± 2.42 a | 125.81 ± 16.63 b | 8.17 ± 0.08 a |
| Alfalfa (M) | 14.63 ± 1.84 c | 5.99 ± 4.00 a | 0.46 ± 0.16 a | 0.48 ± 0.08 b | 18.50 ± 0.62 a | 105.12 ± 9.69 c | 8.23 ± 0.08 a |

Notes: Within the same column, different lowercase letters represent significant differences ($p < 0.05$), while the same lowercase letters represent no significant differences ($p > 0.05$).

### 3.2. Analysis of Soil Bacterial Community Structure in Different Reclaimed Vegetation Types

3.2.1. OTU and Microbial Abundance Analysis

The bacterial V4–V5 regions were sequenced and 1,178,111 valid sequences were obtained. The effective sequences obtained were randomly sampled, the OTU was clustered at the 97% similarity level, and the OTU dilution curve was drawn, as shown in Figure 2, to analyze the microbial community abundance. A total of 239,092 OUT samples were obtained, of which the yellow rose sample was the largest, accounting for 27.22%; the lespedeza and sweet wormwood herb samples accounted for 25.21% and 24.97%, respectively; and the alfalfa sample was the smallest, accounting for 22.53%. The number of bacteria in the soil showed the rule of yellow rose > lespedeza, sweet wormwood herb > alfalfa. With an increase in sequencing quantity, the slope of the rarefaction curve of all sample points gradually rises and finally tends to be flat. It shows that the sampling is reasonable and can truly reflect the diversity of the soil bacterial community.

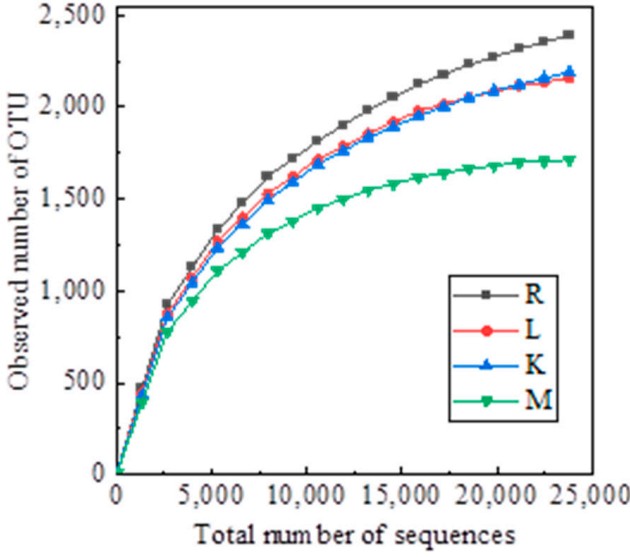

**Figure 2.** Soil bacterial OTU rarefaction curves with different vegetation types.

3.2.2. α-Diversity Analysis

The α-diversity index of the soil microbial community was calculated with 97% similarity OTU (Table 2). The Chao, ACE, and Shannon indexes of the bacterial community diversity show that the yellow rose sample is significantly larger than the lespedeza and sweet wormwood herb samples, and the alfalfa sample is significantly the smallest. In the Shannon index of bacterial community richness and evenness, the significance of the alfalfa samples is low, and the difference between the lespedeza and sweet wormwood herb samples is not significant.

**Table 2.** Microbial community diversity index of reclaimed soils with different vegetation types.

| Diversity Index | Chao Index | ACE Index | Shannon Index |
|---|---|---|---|
| Yellow Rose | 2939.82 ± 243.02 a | 2900.7 ± 215.03 a | 9.36 ± 0.01 a |
| Lespedeza | 2244.71 ± 88.52 b | 2290.8 ± 117.83 b | 9.17 ± 0.01 a |
| Sweet Wormwood Herb | 2472.99 ± 752.06 b | 2569.2 ± 798.91 b | 8.95 ± 0.91 a |
| Alfalfa | 1725.89 ± 711.69 c | 1725.8 ± 712.05 c | 8.30 ± 1.65 b |

Notes: a mean significant difference at 5%, b mean significant difference at 1%, c mean significance difference at 0.1%.

In terms of bacterial community diversity, the soil bacterial community diversity indexes of different plant types also show consistent rule, which is consistent with the research results of Wang Mei and Chen Mengli [35,36]. Many existing studies have shown that the diversity of plants is closely related to the diversity of soil microorganisms [37]. The bacterial community diversity of sample alfalfa is the lowest, mainly because the soil ecosystem has been exporting biomass to the outside for a long time without sufficient supplementation and the biodiversity of the entire ecosystem is not high, resulting in a reduction in the total amount and diversity of soil bacterial community.

### 3.2.3. Analysis of Bacterial Community Structure in Reclaimed Soils

The NCBI database was used for comparison, and the OTU taxonomy information was counted. A total of 140 species, 298 genera, 207 families, 121 orders, 90 classes, and 33 phyla of bacteria were detected. At the phylum level, there are eight bacterial phyla with relative abundance greater than 1% (Figure 3a). Proteobacteria is the dominant bacteria in the alfalfa and sweet wormwood herb samples, accounting for 36.09–43.36% of the total number of microorganisms. Proteobacteria and Actinobacteria are the dominant bacteria in the yellow rose and lespedeza samples, accounting for 53.34–53.39% of the total number of microorganisms. At the class level (Figure 3b), α-alphaproteobacteria and actinobacteria are the dominant bacteria of yellow rose, lespedeza, and sweet wormwood herb. Nonleguminous woody dicotyledonous plants and some nitrogen-fixing bacteria of actinobacteria, such as frankia, can form nodule nitrogen fixation with plants, providing nitrogen sources for the soil ecosystem [38], completing the conversion and utilization of nitrogen between the atmosphere and soil. α-proteobacteria and β-proteobacteria are the dominant bacteria of alfalfa. This is similar to the results of other investigations on the composition of soil microbial community structure in the Loess Plateau [39,40]. The soil microorganism species are diverse, and a large number of soil microorganisms are symbiotic or parasitic with plants, and they even become plant pathogens. In addition, most soil bacteria are closely related to soil nutrient conditions, while the form and content of nutrients provided by plants to soil are different [35,41]. At the same time, plant roots and litter provide a source of decomposition materials for soil bacteria. Therefore, the microbial community structure of reclaimed soils varies significantly with different vegetation types, and soil microorganisms and vegetation form a feedback system of interaction.

### 3.3. Prediction and Analysis of PICRUSt Functions of Different Vegetation Types

### 3.3.1. Kyoto Encyclopedia of Genes and Genomes (KEGG) Primary Function

The data obtained using high-throughput sequencing were compared with the KEGG database, and the PICRUSt method was used for function prediction (Figure 4). The primary functions of soil bacteria mainly include metabolism, genetic information processing, environmental information processing, cellular processes, organismal systems, and human diseases. The abundance of the first three functions accounted for 81.52%. The relative abundance of the metabolic function is 50.30–52.13%. At the primary functional level, there is no significant difference in soil bacterial community functions among the different plant types.

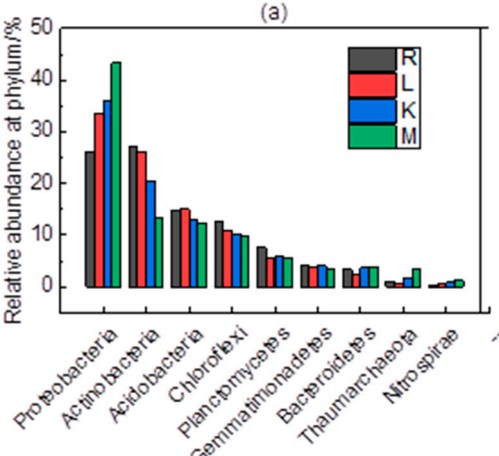 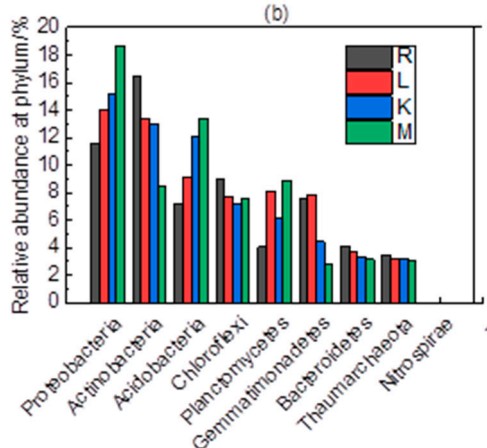

**Figure 3.** Soil bacterial relative abundance (**a**,**b**) in reclaimed soils with different vegetation types. (**a**) phylum. (**b**) class.

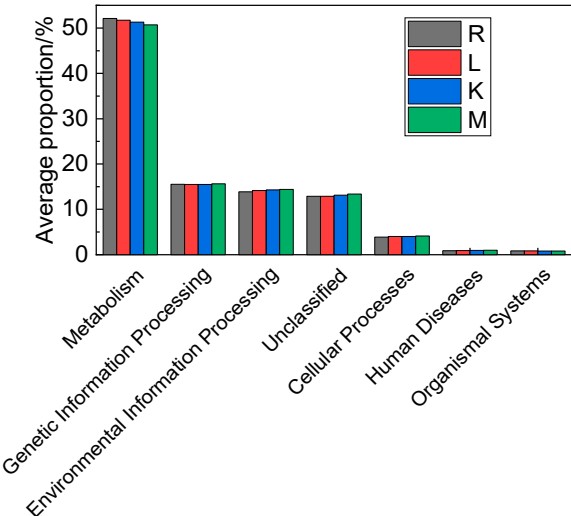

**Figure 4.** Average proportion of KEGG primary functions with different vegetation types.

### 3.3.2. KEGG Secondary Functions

Based on the KEGG data used to predict the secondary functions of soil bacteria, the relative abundance of soil microbial function genes of the different vegetation types was different, and the results are shown in Figure 5.

The functional abundance of carbohydrate metabolism, translation, and energy metabolism is the highest in the alfalfa samples. Comparatively, there is a significant difference among alfalfa, yellow rose, and lespedeza. It indicates that the metabolic capacity of alfalfa soil organic matter is higher. The metabolism of other amino acids, folding, sorting, and degradation and glycan biosynthesis and metabolism have the highest abundance in the yellow rose sample. Therefore, the rhizosphere microorganisms of yellow rose have a high contribution to the decomposition and metabolism of proteins, glycan polypeptides, and amino acids. Signaling molecules and interactions and transport and catabolism functions account for a high proportion in the lespedeza samples, indicating that lespedeza has high rhizosphere microbial activity, a closer relationship between populations, and a more stable community structure. The functions of energy metabolism, carbohydrate metabolism, translation, transportation, and catabolism are relatively significant in sweet wormwood herb samples.

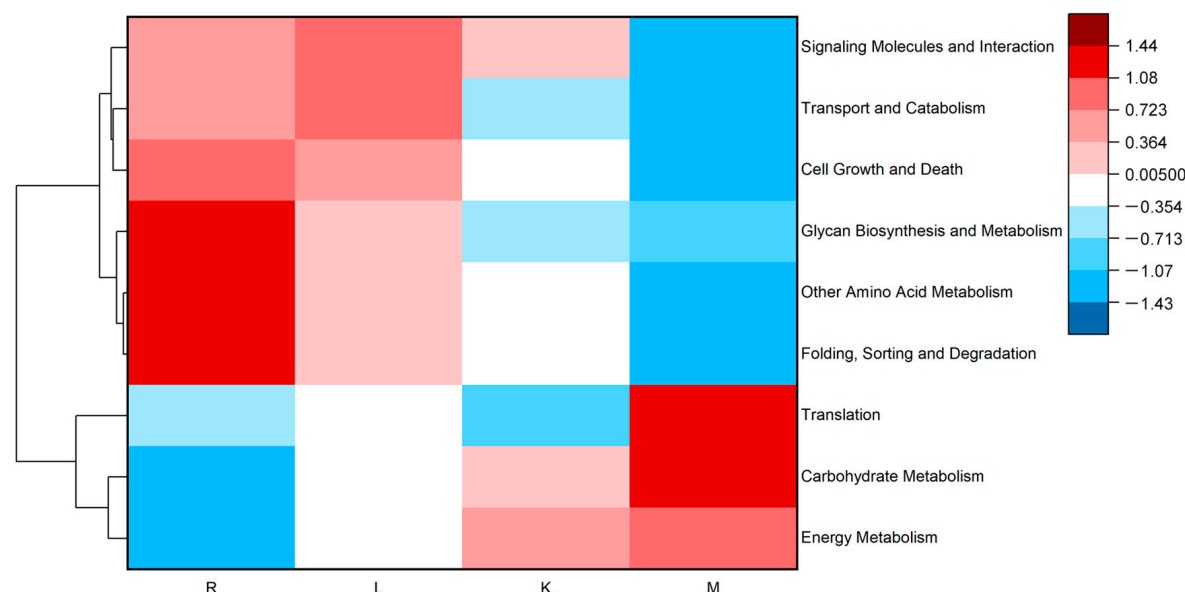

**Figure 5.** Comparison of differences in abundance between KEGG secondary functions with different vegetation types. Notes: M is alfalfa; R is yellow rose; L is lespedeza; K is sweet wormwood herb.

## 4. Discussion

### 4.1. Effects of Soil Environmental Factors on Soil Bacterial Community

Redundancy analysis (RDA) was used to investigate the impact of soil environment on soil bacterial community composition (Table 3 and Figure 6). RDA1 and RDA2 axes respectively explained 46.44% and 35.12% of the difference in bacterial community composition, totaling 81.56%. The RDA sequencing effect between the bacterial community of the sample and different soil environmental factors was good. The RDA1 axis mainly reflects the difference in SOC, and the correlation coefficient is −0.4838; the RDA2 axis mainly reflects the difference in AP and TN, and its correlation coefficients with the ranking axis are 0.6610 and −0.4640, respectively. The long arrow line length of AK, SOC, and TN indicates that the environmental factor is highly correlated with the sample distribution. Therefore, these three factors are factors that affect the soil bacterial community structure, and AK is positively correlated, while SOC and TN are negatively correlated. pH has the least effect on bacterial community structure.

**Table 3.** RDA sorting axis correlation of soil factors with bacterial communities.

| Soil Factors | WC | Temperature | SOC | TN | TP | TK | AK | pH |
|---|---|---|---|---|---|---|---|---|
| RDA1 | 0.2704 | 0.1355 | −0.4838 | −0.3356 | 0.0188 | 0.2474 | 0.4406 | 0.0866 |
| RDA2 | 0.0501 | 0.4314 | −0.4298 | −0.4640 | −0.4357 | −0.1951 | 0.6610 | −0.0485 |

Different plant types have significant effects on the distribution of sample sites. The distribution of sample R and sample L on the RDA2 axis is dispersive, indicating that soil TN and AK, which represent the RDA2 axis, are the main environmental factors. The distribution of samples R and M on the RDA1 axis is dispersive, indicating that SOC representing the RDA1 axis is the main environmental factor. In short, the soil factors affecting the bacterial community are mainly SOC, TN, and AK [42,43].

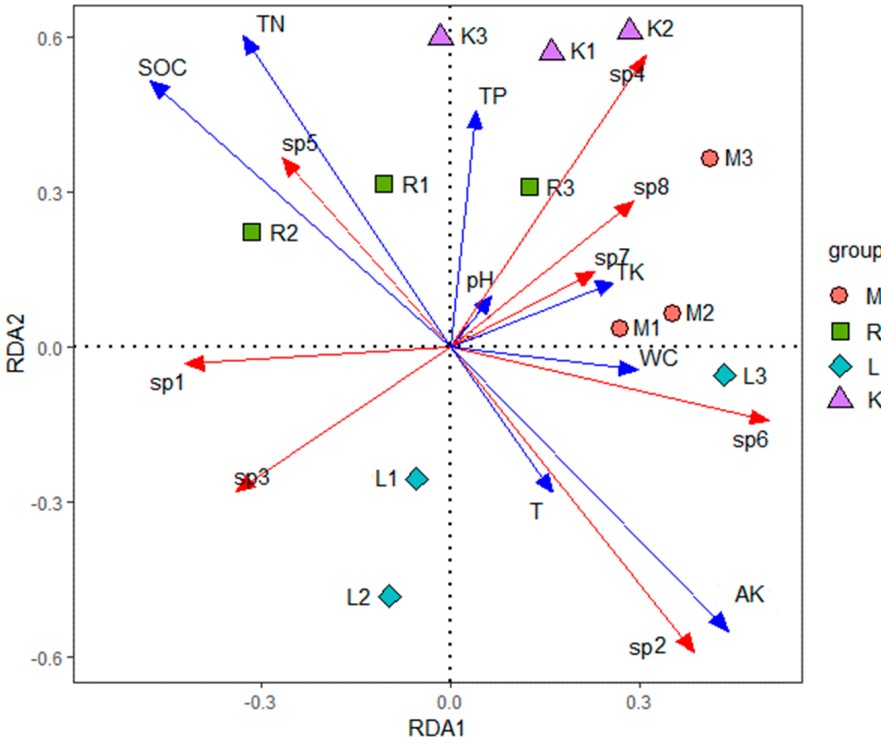

**Figure 6.** Redundancy analysis of bacterial community and soil environmental factors. Notes: sp1, sp2, sp3, sp4, sp5, sp6, sp7, and sp8 stand for alphaproteobacteria, actinobacteria, betaproteobateria, subgroup_6, gammaproteobacteria, thermoleophilia, planctomycetacia, and gemmatimonadetes. Red line stands bacterial community, blue line stands soil environmental factors.

The relative abundance of α-proteobacteria, β-betaproteobateria, and γ-gammaproteobacteria is positively correlated with SOC and negatively correlated with TK. Actinobacteria is positively correlated with AK and negatively correlated with SOC and TN. Thermoleophilia is positively correlated with WC, temperature, and AK, and it was negatively correlated with SOC and TN. Planctomycetacia and gemmatimonadetes were positively correlated with soil WC and TK.

### 4.2. Effect of Soil Environmental Factors on Functional Abundance

In this study, the function prediction and analysis of bacteria in the reclaimed soil show that the soil microorganisms in the study area are mainly involved in category 6 KEGG primary functions and category 41 secondary functions, and the soil microbial community functions are relatively rich. The results show that there is no significant difference in the primary functions layer of soil microbial communities among different plant types, but there are differences in the secondary functional layer. Guixiang et al. found that the relative abundance of subfunctions in Beijing is different in forests and meadows, which may be caused by changes in plant communities, affecting the functional distribution of soil microorganisms [19]. Pearson correlation was used to analyze the correlation between soil bacterial community, environmental factors, and the main secondary functions (Table 4). The metabolism of other amino acids, lipid metabolism, enzyme family, carbohydrate metabolism, and glycan biosynthesis and metabolism were significantly positively correlated with the Chao, ACE, and Shannon indexes of the bacterial community. Energy metabolism function was significantly negatively correlated with the diversity index of the bacterial community. Proteobacteria, as the dominant bacteria in the soil bacterial community, has a very significant negative correlation with secondary subfunctions, such as folding, sorting, and degradation, metabolism of other amino acids, lipid metabolism, enzyme family, cell processes and signaling, carbohydrate metabolism, transport and catabolism, translation, and glycan biosynthesis and metabolism. Folding,

sorting, and degradation, metabolism of other amino acids, cell growth and death, and translation have a significantly positive correlation with actinobacteria. Acidobacteria have seven significant positive correlation functions and two significant negative correlation functions. Planctomycetes have nine significant positive correlation functions and one significant negative correlation function. Chloroflexi and gemmatimonadetes have eight and nine significant positive correlation functions, respectively. Therefore, the abundance of different bacterial species is closely related to functional abundance. The functional abundance of cell communication has a very significant negative correlation with TP and a very significant positive correlation with AK. The soil pH affects the subfunctions of metabolism of other amino acids and translation, which have a very significant negative correlation with proteobacteria. Therefore, it is speculated that soil factors affect their functions by affecting microbial communities. Landesman et al. suggested that soil pH among different vegetation types was an important factor leading to the difference in soil bacterial community in different locations, which further leads to the difference in the functional composition of the soil microbial community [41]. WC, temperature, SOC, TN, and TK have no significant correlation with function.

**Table 4.** Correlation among functional abundance, soil bacterial community, and soil factor properties.

| Soil Functional | Energy Metabolism | Folding, Sorting, and Degradation | Metabolism of Other Amino Acids | Lipid Metabolism | Enzyme Families | Cellular Processes and Signaling | Cell Growth and Death |
|---|---|---|---|---|---|---|---|
| OTU | 0.824 ** | −0.057 | −0.442 | −0.868 ** | −0.833 ** | −0.815 ** | 0.258 |
| Chao index | −0.857 ** | 0.575 | 0.778 ** | 0.730 ** | 0.826 ** | 0.649 * | 0.268 |
| ACE index | −0.836 ** | 0.577 * | 0.776 ** | 0.708 ** | 0.809 ** | 0.629 * | 0.287 |
| Shannon index | −0.911 ** | 0.574 | 0.848 ** | 0.818 ** | 0.935 ** | 0.777 ** | 0.204 |
| Proteobacteria | 0.892 ** | −0.706 * | −0.945 ** | −0.749 ** | −0.910 ** | −0.753 ** | −0.290 |
| Actinobacteria | −0.494 | 0.922 ** | 0.887 ** | 0.242 | 0.465 | 0.174 | 0.827 ** |
| Acidobacteria | −0.920 ** | 0.262 | 0.642 * | 0.929 ** | 0.931 ** | 0.857 ** | −0.106 |
| Chloroflexi | −0.798 ** | 0.557 | 0.855 ** | 0.705 * | 0.860 ** | 0.782 ** | 0.101 |
| Planctomycetes | −0.912 ** | 0.588 * | 0.782 ** | 0.763 ** | 0.852 ** | 0.708 * | 0.244 |
| Gemmatimonadetes | −0.837 ** | 0.607 * | 0.855 ** | 0.714 ** | 0.873 ** | 0.793 ** | 0.125 |
| Bacteroidetes | 0.241 | −0.379 | −0.453 | −0.155 | −0.248 | −0.200 | −0.212 |
| Thaumarchaeota | −0.638 * | 0.028 | 0.350 | 0.685 * | 0.700 * | 0.871 ** | −0.558 |
| Nitrospirae | −0.588 * | 0.062 | 0.413 | 0.639 * | 0.690 * | 0.828 ** | −0.488 |
| WC | 0.213 | 0.068 | −0.095 | −0.173 | −0.159 | −0.242 | 0.138 |
| Temperature | 0.061 | 0.203 | 0.089 | −0.124 | −0.090 | −0.211 | 0.375 |
| SOC | 0.090 | −0.445 | −0.435 | 0.003 | −0.195 | −0.028 | −0.406 |
| TN | −0.049 | −0.386 | −0.348 | 0.138 | −0.063 | 0.066 | −0.385 |
| TP | −0.087 | −0.198 | −0.129 | 0.153 | 0.064 | 0.170 | −0.290 |
| TK | −0.029 | 0.243 | 0.150 | 0.011 | 0.074 | 0.075 | 0.023 |
| AK | −0.146 | 0.409 | 0.295 | 0.014 | 0.109 | −0.133 | 0.580 * |
| pH | −0.439 | 0.566 | 0.693 * | 0.332 | 0.483 | 0.372 | 0.274 |

| Soil Functional | Signaling Molecules and Interaction | Carbohydrate Metabolism | Cell Communication | Xenobiotic Biodegradation and Metabolism | Transport and Catabolism | Translation | Glycan Biosynthesis and Metabolism |
|---|---|---|---|---|---|---|---|
| OTU | 0.775 ** | −0.834 ** | −0.014 | −0.127 | −0.866 ** | −0.201 | −0.842 ** |
| Chao index | −0.408 | 0.783 ** | 0.145 | 0.148 | 0.666 * | 0.598 * | 0.879 ** |
| ACE index | −0.382 | 0.765 ** | 0.159 | 0.147 | 0.644 ** | 0.606 * | 0.861 ** |
| Shannon index | −0.515 | 0.870 ** | 0.003 | 0.261 | 0.792 ** | 0.668 * | 0.939 ** |
| Proteobacteria | 0.451 | −0.830 ** | 0.100 | −0.468 | −0.742 ** | −0.731 ** | −0.925 ** |
| Actinobacteria | 0.192 | 0.317 | 0.127 | 0.446 | 0.181 | 0.985 ** | 0.581 * |
| Acidobacteria | −0.734 ** | 0.909 ** | −0.242 | 0.103 | 0.898 ** | 0.408 | 0.942 ** |
| Chloroflexi | −0.538 | 0.781 ** | −0.299 | 0.623 * | 0.750 ** | 0.559 | 0.846 ** |
| Planctomycetes | −0.471 | 0.817 ** | 0.194 | 0.316 | 0.721 ** | 0.592 * | 0.907 ** |
| Gemmatimonadetes | −0.523 | 0.810 ** | −0.225 | 0.582 * | 0.747 ** | 0.559 | 0.833 ** |
| Bacteroidetes | 0.074 | −0.139 | 0.741 ** | −0.575 | −0.176 | −0.392 | −0.295 |
| Thaumarchaeota | −0.816 ** | 0.784 ** | −0.406 | 0.295 | 0.799 ** | −0.116 | 0.530 |
| Nitrospirae | −0.757 ** | 0.718 ** | −0.603 * | 0.415 | 0.751 ** | −0.039 | 0.531 |
| WC | 0.243 | −0.227 | −0.270 | −0.355 | −0.218 | 0.107 | −0.208 |
| Temperature | 0.324 | −0.175 | −0.287 | −0.097 | −0.162 | 0.405 | −0.070 |
| SOC | −0.211 | −0.097 | −0.255 | −0.144 | −0.013 | −0.527 | −0.140 |
| TN | −0.300 | 0.012 | −0.252 | −0.139 | 0.099 | −0.441 | 0.007 |
| TP | −0.263 | 0.054 | −0.724 ** | 0.102 | 0.173 | −0.181 | 0.028 |
| TK | −0.031 | 0.027 | −0.402 | 0.152 | 0.038 | 0.110 | 0.007 |
| AK | 0.273 | 0.019 | 0.806 ** | −0.002 | −0.077 | 0.475 | 0.242 |
| pH | −0.150 | 0.415 | −0.301 | 0.387 | 0.346 | 0.577 * | 0.496 |

Note: * means significant correlation at $p < 0.05$; ** means significant correlation at $p < 0.01$.

## 5. Conclusions

Many studies have shown that land reclamation and vegetation reconstruction improve soil fertility and soil activity, and they affect the structural composition of the soil microbial community [44,45]. In this study, high-throughput sequencing was used to study the characteristics of rhizosphere soil microbial communities of different types of reclaimed vegetation in the opencast aluminum mine on the Loess Plateau. The results showed that there were significant differences in the structure and function of rhizosphere soil microbial communities among yellow rose, lespedeza, sweet wormwood herb, and alfalfa, and they were closely related to the physical and chemical properties of the soil. The TN and AK contents of the root-soil of the yellow rose were relatively high. The rhizosphere soil had the largest number of microorganisms and the most abundant community structure. The dominant bacteria were proteobacteria and actinobacteria. The functional gene content of the nitrogen metabolism process in the soil was relatively rich. The metabolism of other amino acids, folding, sorting, degradation, glycan biosynthesis, and metabolism are relatively strong and related to the higher TN content of yellow rose. The abundance of soil microbial functional genes involved in the cell activity process in the rhizosphere of lespedeza was high, the diversity of microbial communities was also high, and the relationship between populations was closer, which was related to the TP content of the soil. The SOC content of the alfalfa rhizosphere soil is the highest, and the survey results of functional abundance also show that the SOC content of the alfalfa rhizosphere soil is the highest. The functional abundance of carbohydrate metabolism, translation, and energy metabolism is also the highest, indicating that the metabolic intensity of organic matter is high, thus increasing the nutrient content of the soil, providing nutrients for plants, and further promoting the growth and development of plants. The abundance and diversity of soil microbial communities in the alfalfa soil are lower than those in yellow rose, lespedeza, and sweet wormwood herb. The abundance of actinobacteria in the yellow rose, lespedeza, and sweet wormwood herb soils is higher than that in the alfalfa soil because actinobacteria plays an important role in degrading complex lignin and cellulose and provides nutrients to the soil [39,46]. Therefore, it is advisable to plant meadows such as yellow rose, lespedeza, and sweet wormwood herb to improve the ecological environment of reclaimed soils to improve the soil quality. The differences and diversity of the soil microbial functions should also be considered in the vegetation restoration in the mining area.

PICRUSt technology based on 16S rRNA sequencing data helps ascertain the composition and function of soil bacterial community at the gene level. Because the reclaimed soil of mining wasteland may have the risk of heavy metal pollution and the nutrient is relatively poor, future research on soil microbial functional genes can further focus on the contribution of microorganisms to the decomposition of soil pollutants and the improvement of soil fertility.

**Author Contributions:** All authors made significant contributions to the preparation of this manuscript. Methodology, H.H.; software, D.Z.; conceptualization, H.L.; validation, C.W.; formal analysis, Z.D.; writing—review and editing, J.X. All authors have read and agreed to the published version of the manuscript.

**Funding:** This work was supported by the National Social Science Fund (No. 22BJY064).

**Institutional Review Board Statement:** The study did not require ethical approval.

**Informed Consent Statement:** Not applicable.

**Data Availability Statement:** Not applicable.

**Acknowledgments:** This study was funded by the National Social Science Fund (No. 22BJY064). We thank LetPub (www.letpub.com, accessed on 10 January 2023) for its linguistic assistance during the preparation of this manuscript.

**Conflicts of Interest:** The authors declare that they have no known competing financial interests or personal relationships that could have appeared to influence the work reported in this paper.

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
