# Peer review of "Microbial Community Structure and Predictive Functional Analysis in Reclaimed Soil with Different Vegetation Types: The Example of the Xiaoyi Mine Waste Dump in Shanxi"

_land, doi:10.3390/land12020456_

Round 1
Reviewer 1 Report
This manuscript is well written and organized.
Author Response
We sincerely thank the Reviewers for your enthusiastic work and for your recognition and encouragement of our articles.

Reviewer 2 Report
The authors have carried out interesting work. The organization and arrangement of the manuscript are impressive. However, the manuscript certainly needs to be revised before it can be accepted. Therefore, I recommend a MINOR/MODERATE REVISION.
The following comments should be taken into account with care and love to detail.
1- The abstract section shows 16S rRNA (line: 27). 16S rRNA is also mentioned in the "Materials and Methods" paper, but the method described is not clear enough.
2- The relative abundance of theα-Proteobacteria andβ-Proteobacteria was positively correlated with soil organic carbon (SOC) and negatively correlated with total soil potassium (TK). Actinobacteria was positively correlated with available potassium (AK) and negatively correlated with SOC and total nitrogen (TN). (Lines 37-39). Do these abbreviations follow certain requirements?
3- In the introduction section, the authors are encouraged to cite recent papers describing the importance of different vegetation configurations on soil properties.
4- At the end of the "Introduction" section, could authors 1, 2, and 3 describe the main objectives of this paper? I am not very impressed in terms of novelty and potential impact.
5- Also, the presentation of the results is quite abstract, with very little guidance from the reader through the many algorithms and mathematical details. The authors must improve the clarity of the writing.
6- The "Discussion and conclusion" is poor. It provides a lot of vague and redundant information.
Author Response
Point 1: The abstract section shows 16S rRNA (line: 27). 16S rRNA is also mentioned in the "Materials and Methods" paper, but the method described is not clear enough.
Response 1: Thank you very much for your suggestion, this article really needs to relevant details about 16S rRNA. In lines 193-218 of the revised version, we added relevant operational details about 16s RNA according to the modification. The revised paragraph is shown below:
The 16S rRNA gene of soil bacteria were sequenced within three days of sampling. The DNA of soil samples was extracted using the E.Z.N.A® Soil DNA Kit (Omega Bio-tek, Norcross, GA, USA) according to the manufacturer’s protocols. The V4–V5 area of the 16S rRNA of soil bacteria was amplified by polymerase chain reaction (PCR). The applied degenerate primers included the 515F 5'-barcode-GTGCCAGCMGCCGCGG-3' and 907R 5'–CCGTCAATTCMTTT RAGTTT-3'..
The PCR amplification program was as follows: pre-degeneration at 95°C for 2 min, degeneration at 95°C for 30 s, annealing at 55°C for 30 s, and extension at 72°C for 30 s. The above steps were repeated for 25 cycles, followed by a final extension at 72°C for 5 min. The PCR reaction was performed in triplicate in 20 μL mixtures, containing 4 μL of 5 × FastPfu buffer solution, 2 μL of 2.5 mM dNTP, 0.8 μL of each primer (5μM), 0.4 μL of FastPfu polymerase, and 10 ng template DNA. The amplicons were extracted from 2% agarose gel, and were then purified using the AxyPrep DNA gel extraction kit (Axygen Biosciences, Union City, CA, USA) according to the manufacturer’s instructions. Finally, samples were quantified using QuantiFluor TM-ST (Promega, WI, USA).
Purified PCR products were quantified using the Qubit®3.0 (Promega Corporation, USA) fluorometer, and amplicons with different sequences were mixed evenly. The collected DNA products were then used to establish the Illumina pair-end library with a "Y" shaped connector. An Illumina Nextera® XT Index Kit (Illumina, San Diego, CA) was used to attach dual indices and Illumina sequencing adapters. Following the second PCR, samples were re-cleaned with AMPure XP beads (Beckman Coulter, Pasadena, CA, USA) and quantified. The amplicon library was then pair-end sequenced (2 × 250) using the Illumina MiSeq platform (Shanghai BIOZERON Co., Ltd) according to the standard scheme. The read raw data were then saved in the National Center for Biotechnology Information (NCBI) sequence read archive (SRA) database (Accession Number: SRP05270).
Point 2: The relative abundance of theα-Proteobacteria andβ-Proteobacteria was positively correlated with soil organic carbon (SOC) and negatively correlated with soil total potassium (TK). Actinobacteria was positively correlated with available potassium (AK) and negatively correlated with SOC and total nitrogen (TN). (Lines 37-39). Do these abbreviations follow certain requirements?
Response 2: Thank you for your detailed suggestions, there are some errors in the original manuscript. After correction, the abbreviation of soil organic carbon is SOC, the abbreviation of total potassium is TP, and the abbreviation of total nitrogen is TN. In this thesis, potassium is replaced by kalium, therefore, he abbreviation of available kalium is AK.
Point 3: In the introduction section, the authors are encouraged to cite recent papers describing the importance of different vegetation configurations on soil properties.
Response 3: Thank you very much for your suggestion. We have revised the original manuscript according to the revision, see lines 77-78, 82-84, 97-105 of the revised manuscript.
Point 4: At the end of the "Introduction" section, could authors 1, 2, and 3 describe the main objectives of this paper? I am not very impressed in terms of novelty and potential impact.
Response 4: Your suggestions are much appreciated and we have rewritten the “introduction” section, see lines 120-123 of the revised version:
Based on the above analysis, the characteristics of soil microbial communities under different vegetation restoration were studied in the waste dump of Xiaoyi open aluminum mine pit, Shanxi Province. The structural characteristics and functional sequencing of soil microbial communities were analyzed using the 16S rRNA gene high-throughput sequencing technology and the PICRUSt method. This paper studied from two aspects: 1) Analyze the differences of soil microbial communities structural and functional in different types of vegetation restoration; 2) Analyze the influence among soil properties, soil microbial communities and soil functions. These findings provide a theoretical reference for the selection of reclamation plants and the evaluation of reclamation effects.
Point 5: Also, the presentation of the results is quite abstract, with very little guidance from the reader through the many algorithms and mathematical details. The authors must improve the clarity of the writing.
Response 5: Thank you very much for your suggestion. Based on the comments of the specialists, we have revised the “Results and analysis” section, see section 3.2.2 and section 3.2.3 content (lines 270-314).
Point 6: The "Discussion and conclusion" is poor. It provides a lot of vague and redundant information.
Response 6: We agree with your suggestion. We have reorganized the “Discussion and conclusion” section by adding more details and valid information and removing vague and redundant parts to make the article more logical and scientifically sound. See lines 347-557.

Reviewer 3 Report
In this manuscript, the characteristics and ecological functions of soil bacterial communities in open-pit mines on the Loess Plateau were studied under different vegetation restoration methods. The research object was to reclaim the soil of the aluminum mine waste dump in Xiaoyi County, Shanxi, and 16rRNA gene high-throughput sequencing technology and PICRUSt method were used for specific analysis. This work is significant, but there are some problems that need to be improved. The following comments are intended to improve your manuscript.
1. How to evaluate the good RDA sequencing effect between sample bacterial communities and different soil environmental factors?
2. Why are the Lespedeza bicolor Turcz, Rosa xanthina Lindl and Medicago Sativa Linn all named after the first letter of their English names, while Artemisia annua L is named after K, is there a special meaning?
3. The prediction of soil microorganisms should be the focus of this paper, but the content mentioned is less.
4. The logic of the introduction and discussion is somewhat confused.
5. Table 1, 3 and 4 content is not regular, need to consult other literature of the journal.
6. Table 3 shows a column of data in SOC and TN, which is not clearly expressed
7. There is no SOC in Figure 6, but it is explained.
8. Table format is wrong. It is not a three-line table.
Author Response
Point 1: How to evaluate the good RDA sequencing effect between sample bacterial communities and different soil environmental factors?
Response 1: Thank you for your comments. RDA analysis can reflect the relationship between samples and environmental factors on the same two-dimensional sorting char. The length of the arrow line indicates the degree of correlation. The longer the line, the greater the correlation. The included angle between the arrow line and the sorting axis and the included angle between the arrow line represent the correlation, the acute angle represents the positive correlation, and the obtuse angle represents the negative correlation. The smaller the included angle, the higher the correlation.
Point 2: Why are the Lespedeza bicolor Turcz, Rosa xanthina Lindl and Medicago Sativa Linn all named after the first letter of their English names, while Artemisia annua L is named after K, is there a special meaning?
Response 2: Thank you for your comments. K has no a special meaning, but is the representative of Artemisia annua L.
Point 3: The prediction of soil microorganisms should be the focus of this paper, but the content mentioned is less.
Response 3: Thank you very much for your suggestion. The content of the paper is to predict the soil function using PICRUSt method, and then analyze the function of different vegetation restoration, with the focus on the correlation between soil function and soil quality, soil microorganism, etc. Therefore,in the introduction of this study, the research progress of soil predictive function is added in lines 96-104.
Point 4: The logic of the introduction and discussion is somewhat confused.
Response 4: Thank you very much for your suggestion. We have made changes based on the comments, see lines 76-78, 96-104, 381-419 of the revised version.
Point 5: Table 1, 3 and 4 content is not regular, need to consult other literature of the journal.
Response 5: Thank you very much for your suggestion. We have made changes based on the comments, see Table 1, 3 and 4 of the revised version.
Point 6: Table 3 shows a column of data in SOC and TN, which is not clearly expressed.
Response 6: Thank you very much for your suggestion. We have made changes based on the comments, see Table 3 of the revised version:
Table 3 RDA sorting axis correlation of soil factors with bacterial communities
Soil factors |
WC |
Temperature |
SOC |
TN |
TP |
TK |
AK |
pH |
RDA1 |
0.2704 |
0.1355 |
-0.4838 |
-0.3356 |
0.0188 |
0.2474 |
0.4406 |
0.0866 |
RDA2 |
0.0501 |
0.4314 |
-0.4298 |
-0.4640 |
-0.4357 |
-0.1951 |
0.6610 |
-0.0485 |
Point 7: There is no SOC in Figure 6, but it is explained.
Response 7: Thank you very much for your suggestion. We have modified Figure 6, OC is misspelled, the correct one should be SOC.
Point 8: Table format is wrong. It is not a three-line table.
Response 8: Thank you very much for your suggestion. We have made changes based on the comments, see Table 1, 2 and 3 of the revised version.

Reviewer 4 Report
This is an interesting manuscript. I can recommend it for publication, however, I have a few suggestions that need to be resolved before possible publication:
1. Please write the name of the culture in italics; 2. For such a weighty and large article, I think the 32 publications used are not enough.3. You must also make the strains publicly available to other researchers if they are already patented, ie. by storing them in one of the public microbial collections. Have you included them?
Author Response
Point 1: Please write the name of the culture in italics;
Response 1: Thank you very much for your suggestion, we have re-written the name in italics in this manuscript.
Point 2: For such a weighty and large article, I think the 32 publications used are not enough.
Response 2: We agree with your suggestion. In the revised version, we added a total of 14 references to ensure the richness and scientific content of the article.
Point 3: You must also make the strains publicly available to other researchers if they are already patented, ie. by storing them in one of the public microbial collections. Have you included them?
Response 3: Thank you very much for your suggestion. These bacterial types belong to the classification of soil bacteria, and do not involve patent issues. Therefore, it does not need to be publicly available.

Round 2
Reviewer 2 Report
The author has revised it quite well, and I agree that the paper should be published.
Reviewer 3 Report
The manuscript has been significantly improved and can be accepted for publication.